# The Effect of Dexlansoprazole on Gastroesophageal Reflux Disease: A Systematic Review and Meta-Analysis

**DOI:** 10.3390/ijms25021247

**Published:** 2024-01-19

**Authors:** Gabriel Pereira Nunes, Thayná Cerqueira Silveira, João Vítor Silveira Marciano, Alexandre Henrique dos Reis-Prado, Tulio Morandin Ferrisse, Evandro Barbosa dos Anjos, Maria Helena Fernandes

**Affiliations:** 1Department of Restorative and Preventive Dentistry, School of Dentistry, São Paulo State University (UNESP), Araçatuba 16018-805, Brazil; gabriel.p.nunes@unesp.br; 2Laboratory for Bone Metabolism and Regeneration, Faculty of Dental Medicine, University of Porto, 4160-007 Porto, Portugal; 3Associated Laboratory for Green Chemistry—LAQV/REQUIMTE, University of Porto, 4050-453 Porto, Portugal; 4Department of Medicine, Institute of Health Sciences (ICS), United Colleges of Northern Minas (FUNORTE), Montes Claros 39404-006, Brazil; 5Department of Restorative Dentistry, School of Dentistry, Federal University of Minas Gerais (UFMG), Belo Horizonte 31270-901, Brazil; 6Cariology, Restorative Sciences and Endodontics, School of Dentistry, University of Michigan, Ann Arbor, MI 48104, USA; 7Department of Diagnosis, Surgery and Oral Medicine, School of Dentistry, São Paulo State University (UNESP), Araraquara 14801-385, Brazil; tulio.m.ferrisse@unesp.br

**Keywords:** gastroesophageal reflux, PPI, dexlansoprazole, systematic review, meta-analysis

## Abstract

This systematic review and meta-analysis evaluated the efficacy of dexlansoprazole (a proton pump inhibitor—PPI) in resolving heartburn, reflux, and other symptoms and complications resulting from gastroesophageal reflux disease (GERD). The study followed PRISMA 2020 and was registered in PROSPERO (CRD42020206513). The search strategy used MeSH and free terms appropriately adapted for each database. Only randomized clinical trials (RCTs) were included. The Cochrane tool (RoB 2.0) was used to assess the risk of bias, and the certainty of evidence was rated using GRADE. Ten RCTs were included. Dexlansoprazole outperformed the placebo and other PPIs in the resolution of heartburn and reflux symptoms in patients with GERD, with benefits during and after treatment, especially in those with moderate and severe symptoms. The meta-analyses indicated that dexlansoprazole at doses of 30 and 60 mg had more 24 h heartburn-free days and nights compared to the placebo medications; no difference was reported between dexlansoprazole at doses of 30 and 60 mg in heartburn-free nights. A low bias risk and a moderate certainty of evidence were observed. This review confirms the therapeutic effect of dexlansoprazole (placebo-controlled) and its improvements in GERD symptoms compared to another PPI. However, the interpretation of the results should be carried out cautiously due to the small number of included studies and other reported limitations.

## 1. Introduction

Gastroesophageal reflux disease (GERD) is a common disease of the digestive system with a worldwide prevalence of 8 to 33% [1,2,3] and is characterized by the persistent reflux of gastroduodenal content that ascends through the esophageal tube, causing esophageal and extraesophageal signs and symptoms. In addition to clinical symptoms such as acid regurgitation, heartburn, dental erosions, halitosis, chronic cough, asthma, and recurrent pneumonia [4], GERD also has negative psychological effects on the patient, being responsible for large public health expenditures [3,4,5]. Erosive esophagitis (EE) is a complication of GERD characterized by lesions on the esophageal duct, which is observed in 40% of patients, and its severity can be graded using the Los Angeles (LA) criteria from A to D [6,7,8].

Empirical treatment with proton pump inhibitors (PPIs) presents a sensitivity of 71% and a specificity of 44% for a diagnosis of GERD when compared to other methods such as pH monitoring and upper digestive endoscopy [1,3,9]. Due to their high sensitivity and low cost, PPIs are prescribed for 4 to 8 weeks in a clinical test for patients with typical symptoms and an absence of alarming signs, such as dysphagia, odynophagia, anemia, digestive bleeding, weight loss, a family history of cancer, nausea and vomiting, and an age > 45 years [2,4,5,9]. PPIs decrease the production of hydrochloric acid (HCl) in the stomach lumen, which leads to an increase in gastric pH lasting approximately 24 to 48 h, then reduce the corrosive effects that this content would have on the gastroesophageal walls during reflux [8,10].

However, some limitations of PPIs are related to their ability to inhibit only 70% of receptors at each oral administration, so it takes 2 to 3 days to reach a total inhibition of acid secretion, in addition to the ineffective coverage of most PPIs at night due to their short half-lives and single-dose administration in the morning [11,12]. Therefore, in cases of recurrent nocturnal symptoms, it is common to either double the administration to morning and early evening to increase the drug coverage period or to prescribe a prolonged-release medication, such as dexlansoprazole—an isomer of lansoprazole—which would make a single daily administration sufficient to relieve symptoms [12,13].

The prolonged effect of dexlansoprazole formulation is related to the release of its substrate in pulses according to the pH levels, presenting two peaks of action: the first one occurs in 1 to 2 h at pH levels equal to 5.5 in the proximal duodenum, providing the release of ~25% of the substrate, and the second peak is observed 4 to 5 h after ingesting the drug, corresponding to the release of the remaining 75% with a pH equal to 6.75 in the distal duodenum, ensuring 24 h coverage [13,14,15]. Another advantage of dexlansoprazole compared to other PPIs is its facilitated absorption, i.e., while other drugs need to be taken 30 to 60 min before the first meal of the day to allow for absorption and metabolization, dexlansoprazole does not depend on the time of administration or gastric emptying [13,14,15,16]. Hence, these benefits may lead to improvements in a patient’s quality of life, demonstrating great promise in the treatment of dyspepsia with PPI-class medications [13,14,16].

Thus, given the pharmacological properties of dexlansoprazole in the treatment of GERD, its role in the control of/reduction in symptoms resulting from this disease, and the scarcity of a systematic evaluation of this drug profile, the objective of the current systematic review and meta-analysis was to evaluate the efficacy of dexlansoprazole in the resolution of heartburn, reflux, and other symptoms/complications derived from GERD.

## 2. Material and Methods

### 2.1. Protocol and Registration

This systematic review and meta-analysis was structured according to the Preferred Report Items checklist for Systematic Reviews and Meta-Analyses (PRISMA) [17], taking into account different published models [18,19,20], and following the Cochrane Handbook guidelines [21]. The study was registered in the International Prospective Registry of Systematic Reviews (PROSPERO—CRD 42020206513).

### 2.2. Eligibility Criteria

The inclusion criteria were randomized clinical trials (RCTs) addressing patients with GERD treated with dexlansoprazole, other PPIs, or a placebo. Patients with clinical characteristics, i.e., the presence of heartburn and acid regurgitation as proven by endoscopy, were included. The exclusion criteria considered studies involving patients with the following conditions: on-going medication with anti-secretory agents, such as PPIs and histamine-2 receptor antagonists (H_2_RA), in the 2 weeks prior to an endoscopy; the coexistence of a peptic ulcer or gastrointestinal malignancies; pregnancy; a serious concomitant disease (e.g., decompensated liver cirrhosis and uremia); and previous gastric surgery. Exclusion was also considered for prospective and retrospective studies, case series, case reports, non-human studies, literature review articles, and studies based on research or expert opinions. No language restrictions were applied. A specific question was asked based on the population, intervention, control, and outcome (PICO) criteria: “Is dexlansoprazole effective in resolving heartburn, reflux, and other symptoms derived from gastroesophageal reflux disease?” Considering these criteria, the population was composed of symptomatic patients with GERD. The intervention consisted of a treatment with dexlansoprazole compared to that with another PPI or placebo medication. The outcome was the resolution of heartburn, reflux, and symptoms/complications resulting from GERD.

### 2.3. Databases and Search Strategy

Two independent authors (GPN and TCS) conducted an electronic search of PubMed/MEDLINE, Scopus, Web of Science, Embase, and the Cochrane Library for articles indexed until 25 October 2023. A librarian guided the electronic search strategy, using MeSH terms and free terms appropriately adapted for each database (see Appendix A).

A manual search was performed to identify manuscripts that might not have been retrieved by the electronic search and was also carried out for articles published in the following journals: *Alimentary Pharmacology & Therapeutics, Gastroenterology, Journal of Gastroenterology, Clinical and Experimental Gastroenterology, The American Journal of Gastroenterology, Current Opinion in Gastroenterology, World Journal of Gastroenterology,* and *Drug Design, Development and Therapy.* To find unpublished or ongoing studies, the registration of clinical trials was investigated on the ClinicalTrials.gov website (www.clinicaltrials.gov, accessed on 25 October 2023), with no restriction on the date or language of publication. In addition, the grey literature (produced at government, academic, business, and industrial levels in print or electronic format but not controlled by commercial publishers) was consulted using OpenGrey (http://www.opengrey.eu, accessed on 25 October 2023).

### 2.4. Study Selection and Data Extraction

The study selection was carried out by two reviewers (GPN and TCS) independently in a two-step process. Initially, the authors appraised the titles/abstracts of the studies retrieved from the searches. Studies with titles and abstracts that met the eligibility criteria were included straight away. In Step 2, the authors performed a full-text assessment of the remaining records. Two authors (GPN and TCS) collected the following data from the articles: author/year (location), study design, characteristics of the intervention, number of patients (n), sex, groups, mean age, sample characteristics, use of medication, presence of systemic alterations, method of analysis, monitoring of interventions, and outcomes. Subsequently, a third author (JVS) reviewed the data.

### 2.5. Kappa Analysis

The kappa index was used to calculate the agreement between readers during the process of including publication data. Any disagreements were resolved through discussion and the consensus of all authors.

### 2.6. Bias Risk Assessment

The risk of bias assessment of the included studies was performed by two independent reviewers (GPN and TCS) using the Cochrane Randomized Bias Risk Assessment Tool (RoB 2.0) for the risk of bias analysis (http://handbook.cochrane.org, accessed on 13 August 2023). The evaluation criteria comprised six items: the generation of a random sequence, the concealment of allocations, the blind evaluation of the results, the blindness of the participants and the team, results with incomplete data, selective outcome reporting, and other possible sources of bias. The six domains were evaluated, and the included studies were classified. During the assessment of the risk of bias, any differences between the reviewers were resolved by discussion and consensus and, if necessary, with the help of a third reviewer (JCS).

For each aspect of the quality analysis, the risk of bias for each domain was identified following the recommendations of the Cochrane Handbook for Systematic Reviews of Interventions 5.1.0 (http://handbook.cochrane.org, accessed on 13 August 2023). Each criterion was scored as “yes”, indicating a low risk of bias; “No”, indicating a high risk of bias; or “unclear”, indicating a lack of information or uncertainty about the potential for bias.

### 2.7. Quality of Evidence

The certainty of the evidence (certainty in the effect estimates) was determined for the outcomes using the Grading of Recommendations Assessment, Development, and Evaluation (GRADE) approach. Randomized clinical studies start as strong evidence, and the quality of certainty in the body of evidence decreases to moderate, low, or very low if serious or very serious issues related to the risk of bias, inconsistency, indirectness, imprecision, and publication bias are present [22]. The evaluations were carried out independently by two researchers (GPN and TCS) and then compared.

### 2.8. Synthesis of Results

Studies exhibiting methodological homogeneity were incorporated into the meta-analyses, which were conducted for 24 h heartburn-free days and nights using R software version 3.6.3 with the “META” package. The meta-analyses were based on the Mantel–Haenszel and inverse variance methods. Eight meta-analyses were performed comparing dexlansoprazole at a dose of 30 or 60 mg, or between these doses, with the placebo group. All analyses were measured according to the odds ratio (OR) and a 95% confidence interval (CI). Statistical heterogeneity was assessed using *I*^2^ statistics.

## 3. Results

### 3.1. Bibliographic Search

The database search retrieved 2273 studies from the analyzed sources, as follows: 173 from PubMed/MEDLINE, 269 from Scopus, 430 from Web of Science, 1264 from Embase, 135 from Cochrane Library, and 2 from manual searching. Duplicate studies were removed. After evaluating the titles and abstracts, 16 articles were selected for the eligibility assessment (Figure 1). After reading the full texts of these articles, six articles were excluded [23,24,25,26,27,28] (Table 1), and 10 randomized clinical studies were included (Table 2) [29,30,31,32,33,34,35,36,37,38]. The kappa agreement between investigators for articles that were included in all databases (k = 0.91) presented an acceptable level of agreement.

### 3.2. Study Description

The characteristics of the 10 selected randomized clinical studies are listed in Table 2. In total, 9403 patients with GERD, with an approximate mean age of 41.2 years, were included in the studies, 5814 of whom underwent therapy with dexlansoprazole (intervention), and 3589 were treated as the control groups (placebo: 1002; another PPI: 2587). Regarding the control group, five studies compared dexlansoprazole with a placebo medication [31,34,35,36,37], four compared dexlansoprazole with another PPI [29,30,32,38], and one used both aforementioned controls [33]. The tested dosages of dexlansoprazole were 30, 60, and 90 mg daily. The follow-up period of the interventions ranged from 24 h post-treatment to an evaluation over 6 months.

In all the studies, the patients were clinically screened, and the diagnosis was confirmed with an endoscopic evaluation. Regarding the types of GERD analyzed, two studies evaluated non-erosive GERD (NERD) [34,35], six evaluated GERD with erosive esophagitis [30,31,32,36,37,38], and two assessed both types of GERD [29,33]. To assess the state of erosive esophagitis, studies used the LA classification system, with six articles reporting the intervention in patients in LA grades A, B, C, and D [29,31,33,36,37,38], and two included individuals with LA grades A and B [30,32].

Regarding other systemic conditions, four studies reported the presence of *Helicobacter pylori* infection [30,32,33,35]. However, only the study conducted by Fass et al. [35] mentioned this variable in their outcomes, and they showed that the effects of dexlansoprazole at 30 and 60 mg doses remained significantly greater than the placebo in controlling heartburn over 24 h. They also observed that there were no differences in efficacy between *H. pylori*-positive and H. pylori-negative patients after 4 weeks of treatment with 30 or 60 mg of dexlansoprazole.

Validated questionnaires were also used to assess symptoms derived from GERD, such as: the Gastroesophageal Reflux Questionnaire (GERDQ) [30,31,32]; the Symptom Severity Index (PAGI-SYM) [33,34,35,36,37]; the Disorders Quality-of-Life Index Questionnaire (PAGI-QOL) [36,37]; a questionnaire designed using the reflux symptom index (RSI) [29]; Daily Diaries—The Gastrointestinal Symptom Rating Scale (GSRS) [38]; eDiary entries [31]; the Pittsburgh Sleep Quality Index (PSQI); the Nocturnal Gastroesophageal Reflux Disease Symptom Severity and Impact Questionnaire (N-GSSIQ); and the Work Productivity and Activity Impairment questionnaire (WPAI) [34].

### 3.3. Outcome Results in the Treatment of GERD

Table 2 summarizes the characteristics of the selected studies, including their clinical outcomes. The results were analyzed for the efficacy of dexlansoprazole compared to a placebo or another PPI.

#### 3.3.1. Efficacy of Dexlansoprazole versus Placebo Group

Five studies showed a greater efficacy of dexlansoprazole at a dose of 30, 60, and/or 90 mg in resolving symptoms derived from non-erosive [33,34,35] and erosive GERD [36,37], such as heartburn and regurgitation, compared to a placebo. Conversely, one study found no statistical difference between the dexlansoprazole (30 mg) and the placebo groups; however, the absence of heartburn was 86.6% and 68.1% for these groups, respectively. [31] Regarding the dosages of dexlansoprazole used in erosive GERD, Metz et al. [37] observed that the maintenance rates by month 6 using the life table method were similar (80% and 82%, respectively) in the dexlansoprazole MR 30 mg and 60 mg treatment groups among patients with a baseline grade of A or B. However, for patients with LA grades C and D at baseline, 63% and 85% had maintained a healed EO in the dexlansoprazole MR 30 mg and 60 mg treatment groups, respectively. Nevertheless, Howden et al. [36] showed a similar maintenance of healed erosive esophagitis (Los Angeles A-D) over 6 months between dexlansoprazole MR (60–90 mg) groups.

#### 3.3.2. Efficacy of Dexlansoprazole versus Another PPI

All the studies comparing dexlansoprazole with another PPI [29,30,32,33,38] evaluated the condition of erosive GERD.

A comparison of dexlansoprazole 60 mg and esomeprazole 40 mg was addressed in two studies (patients with EE Los Angeles A, B), and no statistically significant differences were noted in the serial change in the GERDQ score [30,32]. However, one of these trials showed an improvement in the GERDQ score in the observation period (week 8 vs. week 24; *p* < 0.001) in the dexlansoprazole group, whereas no continuous improvement was observed in the esomeprazole group (week 8 vs. week 24; *p* = 0.846) [30]. Moreover, the same study reported that the patients treated with dexlansoprazole presented fewer days with reflux symptoms than those in the esomeprazole group (*p* = 0.008). It is important to report that, in a subgroup analysis performed by Liang et al. [32], the female patients achieved higher complete symptom resolutions in the dexlansoprazole group than in the esomeprazole group on day 3 (38.3% vs. 18.4%, *p* = 0.046). An increasing trend toward a higher complete symptom resolution was observed in the dexlansoprazole group by day 7 (55.3% vs. 36.8%, *p* = 0.09).

Dexlansoprazole at a dose of 60 mg was compared with lansoprazole at a dose of 30 mg in three studies (patients with EE Los Angeles grades A, B, C, and D) [29,33,38]. Two studies showed that dexlansoprazole significantly decreased GERD symptoms compared to lansoprazole [33,38]. Peura et al. [33] found significant results, mainly after four weeks (*p* < 0.05). In contrast, Sharma et al. [38] showed similar results in the two PPI groups; however, the treatment with dexlansoprazole was significantly superior when assessing the cure rate regarding the life table and crude rate analyses (*p* < 0.05). The other study [29] demonstrated improvements after eight weeks of treatment in the dexlansoprazole-treated group in both total typical (93.0% vs. 81.3%, *p* = 0.014) and atypical (67.2% vs. 37.9%, *p* < 0.001) GERD symptoms.

### 3.4. Bias Risk and Quality of Evidence

The risk of bias in the 10 selected studies is shown in Appendix B. According to the Cochrane tool (RoB 2.0), the RCTs demonstrated a low risk of bias for generating a random sequence, except for two that showed vagueness [33,38]. Considering concealment, three studies had a low risk of bias [29,30,32], while the others did not provide clear methodological information (selection bias). For the blinding of participants and staff (performance bias), three studies [32,36,38] did not clearly report blinding of the assessment of the results, and two had a high risk of bias [30,31]. All the studies had a low risk of bias in terms of incomplete results and selective reporting. Concerning other risks, two studies showed a high risk of bias [30,31]. The quality of evidence was classified as moderate. The reasons for each GRADE criterion are described in Table 3.

### 3.5. Meta-Analysis

Three studies were included in the meta-analysis [35,36,37]. Comparisons between 30 mg of dexlansoprazole, 60 mg of dexlansoprazole, and a placebo were carried out with regard to 24 h heartburn-free days and nights (Figure 2 and Figure 3). Individuals using 30 (OR = 0.04, CI = 0.00–0.57, *I*^2^ = 96.4%) and 60 mg (OR = 0.04, CI = 0.01–0.30, *I*^2^ = 96.8%) of dexlansoprazole were 96% less likely to present 24 h heartburn-free days compared to individuals using a placebo (Figure 2A,B). Moreover, 30 mg of dexlansoprazole decreased the number of 24 h heartburn-free days compared to 60 mg (OR = 0.65, CI = 0.48–0.88, *I*^2^ = 0%) (Figure 2D). Publication bias was observed with the trim-and-fill method (Figure 3C). Similarly, a reduction in heartburn events at night was observed when using 30 (OR = 0.09, CI = 0.01–0.96, *I*^2^ = 87.1%) and 60 mg (OR = 0.09, CI = 0.02–0.47, *I*^2^ = 92.1%) of dexlansoprazole compared to a placebo (Figure 3A,B). However, no differences between the doses of 30 and 60 mg were observed for this outcome (OR = 0.74, CI = 0.51–1.08, *I*^2^ = 36.5%) (Figure 3D). Publication bias was demonstrated with the trim-and-fill method (Figure 3C).

## 4. Discussion

To our knowledge, this is the first systematic review and meta-analysis to assess the efficacy of dexlansoprazole in resolving heartburn-type retrosternal pain, regurgitation, and other GERD-derived symptoms. Only prospective randomized clinical studies were selected to compare dexlansoprazole with placebo medications and/or other PPIs, as well as to observe the effects on the most common complication of the disease, which is EE.

In clinical practice, symptoms derived from GERD, such as reflux, are routinely the reasons leading patients to seek treatment. Subjects affected by heartburn are regularly treated without an endoscopic analysis for possible gastric mucosa disease. Clinically, empirical therapy with a PPI is common. In addition, the relief and control of symptoms contribute substantially to improving a patient’s quality of life [39]. The control or resolution of symptoms, such as the absence of heartburn, is a practical criterion for choosing this pharmacological-class drug.

Oral-administered PPIs are acid-labile prodrugs normally formulated with an enteric coating to prevent premature activation and degradation in the gastric acid environment. Upon reaching the small intestine, the enteric coating dissolves, and the drug is absorbed into the blood. Then, the PPIs accumulate selectively in the acid environment of the secretory canaliculi of the meal-activated gastric parietal cells, being rapidly protonated. In this way, the PPIs irreversibly inhibit the enzyme H+, K+ ATPase (the proton pump) on the luminal side of the parietal cell, preventing the secretion of hydrogen ions into the gastric lumen. Because PPIs only bind to active proton pumps, occurring in response to a meal, a preprandial dosing of most compounds is required to match local PPI levels and activate parietal cells. Dexlansoprazole is an enantiomer of lansoprazole that is available in a dual delayed-release formulation that lasts longer than other PPIs. This formulation uses two types of granules with pH-dependent dissolution profiles that release the drug at different times during enteric absorption, i.e., in the proximal duodenum (pH 5.5; ~25%) and in the distal small intestine (pH 6.75; ~75%). This unique pharmacokinetic profile, providing two peak serum concentrations, ensures a higher bioavailability and longer circulation time, allowing for prolonged antisecretory activity. Further, the efficacy of dexlansoprazole is not dependent on meal times, which contributes to better patient compliance [15].

In the present study, the qualitative synthesis of all the included RCTs showed the superior efficacy of dexlansoprazole when compared to a placebo group [31,33,34,35,36,37], regardless of the dose evaluated: 30 mg [31,33,34,35,37], 60 mg [31,33,35,36,37], or 90 mg [36]. Dexlansoprazole was significantly better than the placebo at improving symptoms of heartburn and reflux in symptomatic GERD patients, leading to improved sleep quality and a decreased symptom severity and impact on daily activities.

The meta-analysis also demonstrated a significant decrease in 24 h heartburn-free days and nights in individuals after using dexlansoprazole at doses of 30 mg and 60 mg in comparison to placebo medications [35,36,37]. A reduction in 24 h heartburn-free days was also observed when using dexlansoprazole at a dose of 30 mg compared to dexlansoprazole at a dose of 60 mg. In contrast, no difference was observed when comparing dexlansoprazole at doses of 30 mg and 60 mg for heartburn-free nights. According to the included studies, treatment with dexlansoprazole at different dosages led to significantly greater clinical changes from the baseline period compared to placebos in all the analyzed symptoms. Fass et al. [34] observed better therapeutic gains in moderate–severe and severe–very severe heartburn than in milder clinical conditions, registering average differences from the placebo of 32.6 to 65.6% for asymptomatic nights. In contrast, Metz et al. [37] reported a dose equivalence (30 mg and 60 mg) for the typical days free from heartburn, exceeding the placebo formulations by up to 67%. However, when evaluating dexlansoprazole at a dose of 30 mg against EE, there was a drop in its effectiveness among patients with more severe LA stratification. Thus, the relief of heartburn was 80% in the less severe cases A and B but lower in the most severe cases of LA classes C and D (with the improvement dropping to 63%), while the 60 mg formulation maintained 85% relief at the different levels. Howden et al. [36] did not observe differences in efficacy between the 60 mg and 90 mg dosages in the control of EE symptoms. It is worth mentioning that the outcomes reported by patients in GERD/EE studies tend to be less objective, introducing an additional degree of uncertainty into the study results [40]. This probably explains the observation of a higher rate of symptom control for dexlansoprazole at a dose of 30 mg than dexlansoprazole at a dose of 60 mg versus a placebo among the NERD patients reported by Fass et al. [35], as well as the absence of a statistical difference when comparing the dosages of 60 and 90 mg [36].

A quantitative analysis comparing dexlansoprazole with other PPIs was not possible due to the increased methodological heterogeneity regarding the evaluated population, intervention groups, and follow-up period between the selected studies. Chiang et al. [30] showed a lack of a significant difference between dexlansoprazole at a dose of 60 mg and esomeprazole at a dose of 40 mg regarding heartburn; however, they found more intense acid reflux in the group using esomeprazole at a dose of 40 mg (3.3 ± 0.6 vs. 3.0 ± 0.5; *p* = 0.011). These results are in line with those reported by Liang et al. [32]. The great observation shown in these studies is the comparison of dexlansoprazole with a very popular PPI, considered the first option among all others in its class, due to its low cost and high efficiency. In addition, these studies were pioneers in evaluating these PPIs in cases of EE with LA grades A and B. Nevertheless, according to the recent Lyon Consensus, the new diagnostic criteria for GERD do not include LA grades A and B due to unproven GERD [41]. Regarding the new criteria, PPI therapy should be performed in individuals with proven GERD, such as those with grade C or D, which may lead to a misinterpretation of the results of those eligible RCTs using PPI therapy in cases of low-grade esophagitis. Moreover, another limitation of these studies is related to the small number of patients included. Thus, the reduced number of cases may have hindered the observation of a statistical difference. Consequently, dexlansoprazole may be a more suitable single-daily-dose PPI than esomeprazole for use on demand. The advantage of dexlansoprazole is that it employs a new approach, whereby its delayed double-release formulation prolongs the plasma concentration and ultimately extends the duration of acid suppression [42], thus offering a dosing effect twice a day in one single dose.

When a direct comparison was performed with another PPI (lansoprazole), after reassessments, Peura et al. [33] reported greater improvements in heartburn associated or not with reflux in patients using dexlansoprazole at a dose of 60 mg than in those using lansoprazole at a dose of 30 mg (*p* < 0.05). These findings were also corroborated by Lin et al. [29]. Moreover, Sharma et al. [38] performed two evaluations with a large number of patients endoscopically diagnosed with GERD and EE, and the 60 mg and 90 mg dosages of dexlansoprazole achieved better positive results than lansoprazole in both analyses. In the eighth week of therapy, patients with EE Los Angeles grades C-D demonstrated more progress in the group using dexlansoprazole at a dose of 90 mg, but the results for the 60 mg version were similar to those for lansoprazole. This can be justified because dexlansoprazole was formulated to provide a substantially larger area under the curve and prolonged plasma drug levels compared to a conventional release of a PPI [38]. Furthermore, the 60 and 90 mg doses of dexlansoprazole were well tolerated in the studies [36,38], without dose-dependent adverse events and with a side effect profile similar to that of lansoprazole at a dose of 30 mg, for which there are data on the history of its long-term use in patients with GERD and other acid-related disorders [43,44]. The side effects seen with dexlansoprazole are similar in character and severity to those seen with other PPIs [45,46,47]. Although current randomized controlled trials may not have adequate power to demonstrate very rare side effects, those with a large sample size, as well as the strict monitoring of patients, suggest that the compound has an excellent safety profile.

Although the results presented in this systematic review confirm the therapeutic effect of dexlansoprazole (placebo-controlled) and its improvements in GERD symptoms compared to another PPI, they are still insufficient if the health profession’s goal is to choose dexlansoprazole as the first option among other drugs of the same class. Moreover, other limiting factors are the small number of patients being tested and the lack of more accurate tests for the final evaluation, including only questionnaires, such as the PAGI-QOL, PAGI-SYM, PSQI, N-GSSIQ, and WPAI. These methods include subjective views that the patients have about their clinical framework and the repercussions on their daily lives, having little impact on the organic assessment of the disease. Imaging exams were also applied, such as an upper digestive endoscopy, which lacks the objectivity of an esophageal pH-metry—taken as the gold standard for the diagnosis of GERD—and which would also be able to quantify acid reflux, providing valuable information regarding the efficiency of the PPI in inhibiting HCl secretion.

In this sense, in situations of difficult control, dexlansoprazole can influence an improvement in a patient’s quality of life. However, the synthesis of knowledge indicates the need to evaluate the available strategies used in the treatment of GERD in order to compare the effectiveness of the different PPIs to guide the rational prescription of these drugs in the unique conditions presented by each patient. Furthermore, when choosing a PPI, the superiority of dexlansoprazole versus esomeprazole/lansoprazole in the control of symptoms in the treatment of GERD must be considered together with other factors, for example, treatment cost, its interaction with food, and other characteristics.

It is pertinent to mention that food can reduce or compromise the bioavailability of some PPIs; plasma exposure to dexlansoprazole has been shown to increase after the administration of this drug under feeding conditions compared to in a fasting state, with no relevant differences observed in the intragastric pH profile in feeding or fasting states [48]. As low adherence is the main cause of treatment failure [49,50], the ability to administer dexlansoprazole with or without food offers a convenience not available with PPIs that exhibit a negative feeding effect and may improve the treatment effectiveness of this PPI in the current social setting of a patient.

The data from this systematic review and meta-analysis should be carefully analyzed and used to guide the need for further studies in this area. It is noteworthy that the patients in the included studies were carefully selected and allowed to participate based on specific inclusion and exclusion criteria, an approach that is not consistent with a clinical environment. These limitations are common to most clinical studies and should not affect the general applicability of the trial results. It is not clear whether the results of the current studies are generalizable to other groups of patients with GERD excluded from these studies, including those with Barrett’s esophagus and stricture. However, dexlansoprazole showed clinical and statistical advantages over placebos and other PPIs in the assessed population. Thus, it is advised that robust RCT studies be developed, with the inclusion of larger population groups, more accurate efficiency assessment methods, and other methodological approaches, in addition to those already carried out by the studies presented herein, such as, for example, pH monitoring, questionnaires, and upper digestive endoscopies.

## 5. Conclusions

The evidence points out that dexlansoprazole has a satisfactory effect on the resolution of heartburn and reflux symptoms in patients with GERD and its erosive complications, with benefits during and after treatment, in different dosages, and, mainly, in those with moderate and severe symptoms. However, these results should be interpreted with caution due to the small number of included studies and the other reported limitations.

## Figures and Tables

**Figure 1 ijms-25-01247-f001:**
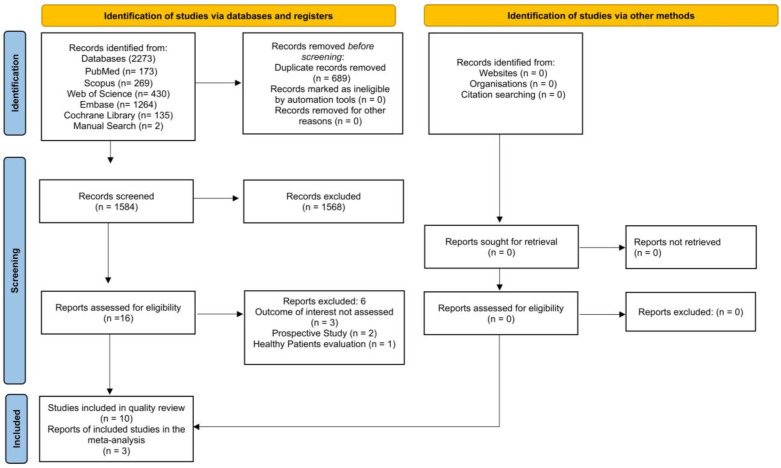
PRISMA flowchart of study selection, showing the number of studies identified, selected, eligible, and included in the review.

**Figure 2 ijms-25-01247-f002:**
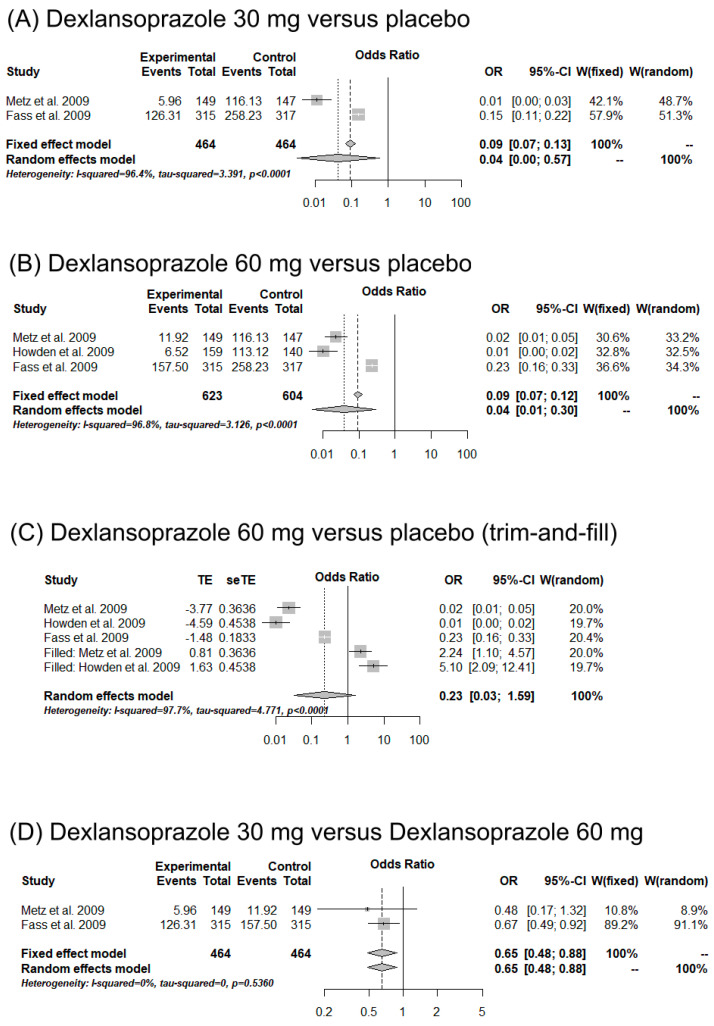
Forest plot of meta-analysis for the studies evaluating 30 mg of dexlansoprazole (**A**), 60 mg of dexlansoprazole (**B**), and trim-and-fill analysis for 60 mg of dexlansoprazole (**C**) compared to placebo medications, and 30 mg of dexlansoprazole compared to 60 mg of dexlansoprazole (**D**) for the outcome event “24 h heartburn-free days” [35,36,37].

**Figure 3 ijms-25-01247-f003:**
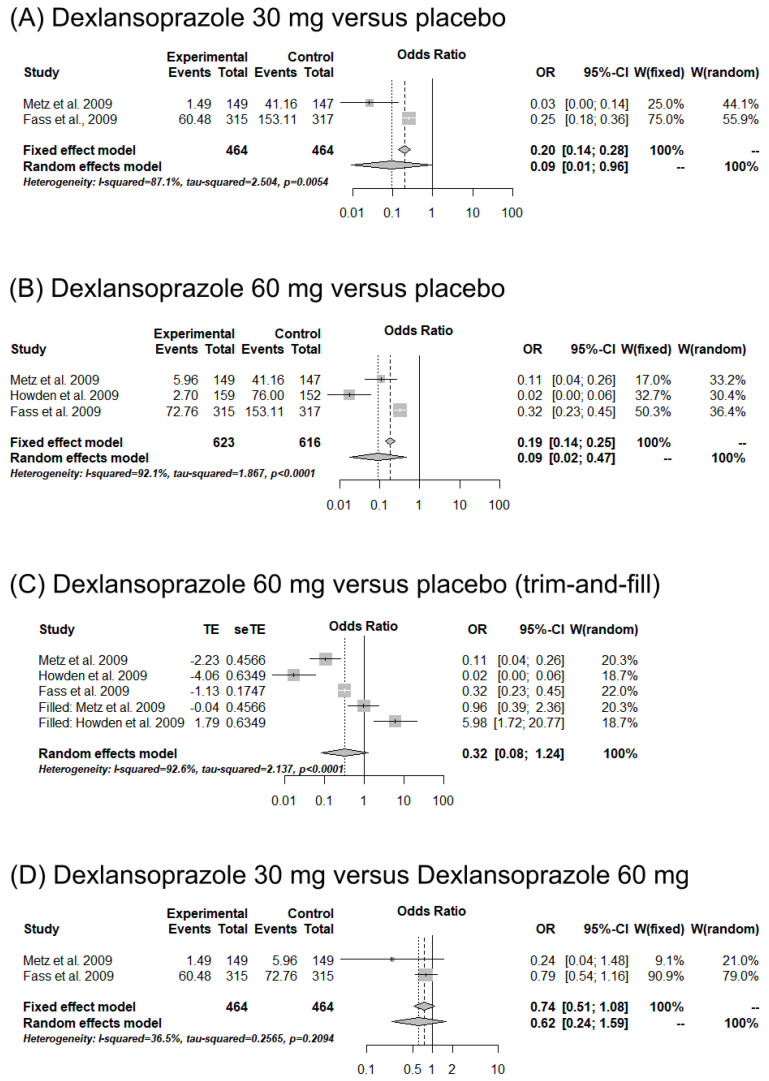
Forest plot of meta-analysis for the studies evaluating 30 mg of dexlansoprazole (**A**), 60 mg of dexlansoprazole (**B**), trim-and-fill analysis for 60 mg of dexlansoprazole (**C**) compared to placebo medications, and 30 mg of dexlansoprazole compared to 60 mg of dexlansoprazole (**D**) for the outcome event “heartburn-free nights” [35,36,37].

**Table 1 ijms-25-01247-t001:** Reasons for the exclusion of the articles.

Reasons	References
Prospective Study	Fass et al., 2012 [26] and Gold et al., 2017 [24]
Outcome of interest not assessed	Peura et al., 2009 [27], Zhang et al., 2009 [28], and Peura et al., 2013 [25]
Healthy patients (absence of GERD)	Han et al., 2023 [23]

**Table 2 ijms-25-01247-t002:** General characteristics of the included studies.

Authors, Year (Local)	G1: Dexlansoprazole G2: Comparison Group (Placebo or Another PPI) Number of Patients (n), and Sex (M/F):	Mean Age	Characteristics of the Samples	Use of Medication and Systemic Disturbs	Method of Analysis	Follow-Up	Outcomes
Lin et al., 2020 [29] (Taipei, Taiwan)	G1: Dexlansoprazole 60 mg; n = 116 (M: 44; F: 67) G2: Lansoprazole 30 mg; n = 116 (M: 36; F: 75)	G1: 58.1 ± 12.4 G2: 56.4 ± 11.5	n (%) Acid regurgitation G1: 64 (55.2%) G2: 76 (65.5%) Erosive esophagitis G1: 28 (24.1%) G2: 32 (27.6%) Los Angeles Score: grades A, B, C, D. Heartburn G1: 52 (44.8%) G2: 47 (40.5%) Typical symptom G1: 100 (86.2%) G2: 107 (92.2%) Cough G1: 51 (43.9%) G2: 50 (43.1%) Globus G1: 66 (56.9%) G2: 78 (67.2%) Hoarseness G1: 21 (18.1%) G2: 20 (17.2%)	Diabetes and Dyslipidemia	Endoscopy and questionnaire designed using reflux symptom index (RSI)	8 weeks	Therapeutic response rate Typical symptoms, n/N (%) G1: 93/100 (93.0%) G2: 87/107 (81.3%) *p* = 0.014 Acid regurgitation, n/N (%) G1: 56/64 (87.5%) G2: 62/76 (81.6%) Heartburn, n/N (%) G1: 48/52 (92.3%) G2: 35/47 (74.5%) *p* = 0.027 Atypical symptom, n/N (%) G1: 78/116 (67.2%) G2: 44/116 (37.9%) < 0.001 Cough, n/N (%) G1: 39/51 (76.5%) G2: 19/50 (38.0%) < 0.001 Globus, n/N (%) G1: 46/66 (69.7%) G2: 24/78 (30.8%) < 0.001 Hoarseness, n/N (%) G1: 11/21 (52.4%) G2: 6/20 (30.0%)
Chiang et al., 2019 [30] (Kaohsiung, Taiwan)	G1: Dexlansoprazole 60 mg; n = 43 (M: 20; F: 23) G2: Esomeprazole 40 mg; n = 43 (M: 20; F23)	G1: 46.9 G2: 50.5	G1: GERD Los Angeles grades A and B G2: GERD Los Angeles grades A and B	n (%) Smoking G1: 9 (22.5) G2: 5 (12.2) Alcohol use G1: 14 (35.0) G2: 15 (36.6) Metabolic Syndrome G1: 24 (60.0) G2: 23(56.1) PPI dependence G1: 33 (82,5) G2: 38 (92,7) *H. pylori* infection G1: 6 (15.0) G2: 6 (14.6)	Endoscopy, Gastroesophageal Reflux Questionnaire (GERDQ)	Clinical GERDQ at 4, 8, 12, 16, 20 weeks and endoscopy at 24 weeks.	Changes in GERDQ scores G1/G2 (Mean ± SD) Week 0: 23.2 ± 3.7/23.7 ± 4.7 Week 4: 17.1 ± 3.7/18.0 ± 4.1 Week 8: 16.4 ± 3.6/16.9 ± 3.7 Week 12: 16.3 ± 4.0/17.4 ± 4.7 Week 16: 14.7 ± 4.4/16.2 ± 4.7 Week 20: 13.7 ± 3.2/15.0 ± 4.8 Week 24: 13.1 ± 3.8/16.5 ± 10.9 Days to symptom resolution G1: 9.2 ± 14.4/G2: 10.5 ± 16.2 *p* = 0.700 Improvement in the GERDQ score in the on-demand period (8 vs. 24 week) G1: *p* < 0.0001/G2: *p* = 0.846 Number of days with reflux symptoms: G1: 37.3 ± 37.8 / G2: 53.9 ± 54.2; *p* = 0.008
Gremse et al., 2019 [31] (USA, Poland, Mexico and Portugal)	G1: Dexlansoprazole MR 30 mg; n = 25 (M: 14; F:11) Dexlansoprazole MR 60 mg; n = 62 (M: 38; F: 24) G2: Placebo; n = 26 (M:16; F:10)	G1: 30 mg: 14.6 60 mg: 14.8 G2: 14.8	Patients with GERD and esophagitis (Los Angeles Score) severity grades A, B, C, or D: G1.1: 30 mg: A (14), B (11), C (0), D (0) G1.2: 60 mg: A (34), B (26), C (1) e D (1) G2: A (16), B (9), C (1), D (0)	Smoker G1.1: 1 (4.0) G1.2: 1 (1.6)	Endoscopy, eDiary entry, and investigator assessment of GERD	Treatment follow-up 3 months later.	EE Healing Phase After 8 weeks of treatment G1.2: 88% EE Healing Phase After 16-weeks, double-blind G1.1: 82%/G2: 55% Absence of Heartburn G1:86.6% of days G2: 68.1% of days Treatment-Free Follow-up Phase: 3 months: Absence of heartburn on average days G1: 86.3% G2: 83.6% Absence of rescue medication G1: 99.1% of days G2: 97.7% of days
Liang et al., 2017 [32] (Taiwan)	G1: Dexlansoprazole 60 mg; n = 81 (M: 34; F: 47) G2: Esomeprazole 40 mg; n = 81 (M: 43; F: 38)	G1: 50.6 ± 13.3 G2: 49.9 ± 12.8	Patients with clinical symptoms of acid regurgitation, heartburn, and a feeling of acidity in the stomach and who had endoscopy-confirmed LA grades A or B erosive esophagitis	N (%) H. pylori infection Previous history G1:10 (12.3) G2:15 (18.5) Current infection G1:10 (12.3) G2: 12 (14.8) Hiatal hernia G1:10 (12.3) G2: 15 (18.5) GEFV (grade 3 or 4) G1: 7 (8.6) G2: 8 (9.9) Esophagitis grade B G1: 15 (18.5) G2: 13 (16.0)	Endoscopy, GERDQ	The study continued for 1 week with evaluations on days 1, 3 and 7.	n (%) G1/G2 CSR Day 1: 21 (25.9)/23 (28.4) CSR Day 3: 27 (33.3)/26 (32.1) CSR Day 7: 42 (51.9)/39 (48.1) Night reflux: 45 (76.3)/40 (74.1) Night heart burn: 20 (33.9)/18 (33.3) Night acid reflux: 20 (33.9)/19 (35.2) Frequency of night symptoms: 2.7 ± 2.0/2.7 ± 2.4
Peura et al., 2013 [33] (Virginia, USA)	GERD non-erosive G1: G1.1: 30 mg (n = 217); G1.2: 60 mg (n = 225) n = 442 (M: 130; F: 312) G2: Palacebo; n = 219 (M: 54; F: 165) EE G1: Dexlansoprazole 60 mg; n = 925 (M: 478; F: 447) G2: Lansoprazole 30 mg; n = 984 (M:507; F: 477)	GERD G1: Dexlansoprazole 30 mg (48.0) 60 mg (46.5) G2: Placebo (46.3) EE G1: Dexlansoprazole 60 mg (48.3) G2: Lansoprazole 30 mg (46.8)	2 studies were performed: the first with patients with a history of heartburn for ≥6 months and a diagnosis of GERD experiencing heartburn ≥4 of 7 days, and the second with endoscopically confirmed EE (LA grades A, B, C, and D): G1: A (283), B (348), C (235) e D (59) G2: A (303), B (378), C (239) e C (64)	GERD: n (%) Smoker G1.1: 43 (19.8) G1.2: 37 (16.4) G2: 40 (18.3) Alcohol drinker G1.1: 105 (48.4) G1.2: 129 (57.3) G2: 131 (59.8) Positive H. pylori G1.1: 67 (30.9) G1.2: 64 (28.4) G2: 64 (29.2) EE: n (%) Smoker G1:234 (25.3) G2: 258 (26.2) Alcohol drinker G1: 534 (57.7) G2: 528 (53.7) Positive H. pylori G1: 9 (1.0) G2: 11 (1.1)	Endoscopy, Symptom Severity Index (PAGI-SYM)	Evaluation performed in weeks 2, 4, and 8.	Mean PAGI-SYM NERD Baseline/week 2/week 4 Heartburn/regurgitation subscale G1.1: 2.66/1.20/0.92 G1.2: 2.68/1.07/0.87 G2: 2.71/1.74/1.50 *p* ≤ 0.00001 Heartburn only G1.1: 3.12/1.40/1.03 G1.2: 3.18/1.29/1.02 G2: 3.16/2.18/1.85 Regurgitation only G1.1: 2.61/1.16/0.93 G1.2: 2.64/1.01/0.85 G2: 2.63/1.62/1.46 Mean PAGI-SYM EE Baseline/week 4/week 8 Heartburn/regurgitation subscale G1: 2.71/0.69/0.56 G2: 2.64/0.76/0.67 *p* < 0.05 Heartburn only G1: 3.30/0.79/0.68 G2: 3.23/0.92/0.81 Regurgitation only G1: 2.63/0.68/0.55 G2: 2.56/0.73/0.65
Fass et al., 2011 [34] (Arizona, USA)	G1: Dexlansoprazole MR 30 mg; n = 152 (M:55; F: 97) G2: Placebo; n = 153 (M: 55; F: 98)	G1: 44.6 ± 11.29 G2: 43.9 ± 12.45	Patients with a history of symptomatic GERD with or without a history of erosive esophagitis diagnosed >6 months before screening	PPI use within 6 months of randomization G1:79 (52.0) G2:79 (51.6) Alcohol drinker G1: 86 (56.6) G2: 81 (52.9) Smoker G1: 35 (23.0) G2: 42 (27.5)	Endoscopy, PSQI Pittsburgh Sleep Quality Index (PSQI), Nocturnal Gastroesophageal Reflux Disease Symptom Severity and Impact Questionnaire (N-GSSIQ), Work Productivity and Activity Impairment (WPAI)	Endoscopy evaluation 4 days before day 1. Period of study therapy: 4 weeks. Evaluations on day 1 and week 4.	(%) Nights without heartburn G1: 73.1/G2: 35.7 Patients with relief of nocturnal heartburn during the last 7 days of treatment: G1: 47.5/G2: 19.6 Patients with relief of GERD-related sleep disturbances during the last 7 days of treatment G1: 69.7/G2: 47.9 Baseline/4 week/changes (mean and SD): Nocturnal GERD symptom severity subscale mean and SD Baseline/4 week/changes G1: 29.20 ± 12.13/10.85 ± 13.03/ −18.35 ± 13.51 G2: 30.33 ± 11.29/18.45 ± 14.67/ −11.88 ± 13.01
Fass et al., 2009 [35] (Arizona, USA)	G1: n = 630 (M: 190; F: 440) G1.1: Dexlansoprazole MR 30 mg; n = 315 (M: 84; F: 231) G1.2: Dexlansoprazole MR 60 mg; n = 315 (M: 106; F: 209) G2: Placebo; n = 317 (M: 84; F: 233)	G1: 47.6 (13.6) G2: 47.6 (14.4)	Patients with NERD who displayed normal mucosa (no EO) at the screening endoscopy.	BMI G1: 627/G2: 317 *Helicobacter pylori* status, n Positive G1: 185/G2: 89 Alcohol use, n Drinker G1: 343/G2: 182 Smoking status, n Smoker G1: 129/G2: 52	Endoscopy, PAGI-SYM	After 4 weeks of self-administration of the drug, all patients were examined and submitted to laboratory evaluations.	Median percentage of 24 h heartburn-free days G1.1: 54.9 G1.2: 50.0 G2: 18.54 Median percentage of nights without heartburn G1.1: 80.8 G1.2: 76.9 G2: 51.7
Howden et al., 2009 [36] (Illinois, USA)	G1: n = 311 (M: 165; F: 146); G1.1 Dexlansoprazole MR 60 mg 159 (M: 83; F: 76) G1.2 Dexlansoprazole MR 90 mg 152 (M: 82; F:70) G2: Placebo; n = 140 (M:70; F: 70)	G1: G1.1—60 mg (49.7) G1.2—90 mg (48.8) G2: 48.2	Patients with erosive esophagitis (EO severity by LA classification—A, B, C, D): G1: A (114), B (119), C (65) e D (13) G2: A (58), B (48), C (28) e D (6)	None	Endoscopy, PAGI-SYM, Disorders Quality-of-Life Index Questionnaire (PAGI-QOL)	Endoscopy before the therapy and 4 or 8 weeks after the therapy. Return visits after 1, 3, and 6 months.	Median days without heartburn during treatment, % 24 h days/Nights G1.1: 95.8/98.3; G1.2: 94.4/97.1; G2: 19.2/50.0 Median of mean severity of heartburn during treatment, % 24 h days/Nights G1.1: 0.03/0.02; G1.2: 0.04/0.04; G2: 1.00/0.83 Median days without rescue medication during treatment, % G1.1: 94.9/G1.2: 93.6/G2: 27.5
Metz et al., 2009 [37] (75 Centers in USA, 19 in non-USA sites)	G1: n = 298 (M: 143; F: 155); G1.1 Dexlansoprazole 30 mg; G1.2: Dexlansoprazole MR 60 mg G2: Placebo; n = 147 (M:72; F: 75)	G1: 30 mg: 47.1 60 mg: 47.9 G2: 49.5	Patients with GERD and EE: EO severity by LA classification—A, B, C, D: G1: A (109), B (103), C (70) e D (16) G2: A (51), B (57), C (34) e D (5)	None	Endoscopy, PAGI-QOL, PAGI-SYM	Outcomes were recorded on day 1 and at months 1, 3, and 6.	Median percentage of 24 h heartburn-free days and median percentage of nights without heartburn during treatment: 24 h heartburn-free days G1.1: 96%/G1.2: 91%/G2: 29% Nights without heartburn G1.1: 99%/G1.2: 96%/G2: 72%
Sharma et al., 2009 [38] (Kansas, USA)	Study 1: G1: n = 1348 (M: 746; F: 602); G1.1 Dexlansoprazole MR 60 mg G1.2 Dexlansoprazole MR 90 mg G2: n = 690 (M: 365; F: 237); Lansoprazole MR 30 mg Study 2: G1: n = 1381 (M: 733; F: 648); Dexlansoprazole MR 60 mg G1.2 Dexlansoprazole MR 90 mg G2: Lansoprazole MR 30 mg; n = 673 (M: 362; F: 275)	Study 1 G1: 60 mg (47.8) 90 mg (47.3) G2: 47.3 Study 2 G1: 60 mg (48.7) 90 mg (47.7) G2: (47.3)	Adult patients’ EO (EO severity by LA classification—A, B, C, D): Study 1 G1: A (478), B (480), C (311) e D (78) G2: A (231), B (248), C (170) e D (40) Study 2 G1: A (505), B (478), C (308) e D (88) G2: A (222), B (257), C (150) e D (44)	None	Endoscopy, patient responses in Daily Diaries—The Gastro Symptom Rating Scale (GSRS)	The screening period lasted up to 21 days. On the week 4 visit and on the week 8 or final visit (if not healed by week 4).	(%) 24 h heartburn-free days: Study 1: G1.1: 82.1/G1.2: 84.2/G2: 80.0; *p* > 0.05 Study 2: G1.1: 83.0/G1.2: 80.8/G2: 78.3 Patients with complete EE healing by week 8 Life table/crude rate Study 1 G1.1: 92.3/85.3—G1.2: 92.2/85.8—G2: 86.1/79.0; *p* > 0.025 Study 2 G1.1: 93.1/86.9; G1.2: 94.9/89.4; G2: 91.5/84.6; *p* < 0.05 Patients with baseline grades C or D EE-healed by week 8 Life table/crude rate Study 1: G1.1: 88.9/79.7—G1.2: 83.8/74.1; G2: 74.5/65.0 Study 2: G1.1: 87.6/77.8; G1.2: 93.3/86.3; G2: 87.7/78.9

PPI: proton pump inhibitor; CSR: complete symptom resolution; GERD: gastro-esophageal reflux disease; EE: erosive esophagitis; LA: Los Angeles grade; GEFV: gastroesophageal flap valve.

**Table 3 ijms-25-01247-t003:** Quality of evidence: Grading of Recommendation, Assessment, Development, and Evaluation (GRADE) instrument.

Quality Assessment							
Nº of Studies	Study Design	Risk of Bias	Inconsistency	Indirectness	Imprecision	Other Considerations	Certainty
10	Randomized trials	Not serious	Serious ^a,b^	Not serious	Not serious	All plausible residual confounding variables would reduce the demonstrated effect.	⨁⨁⨁◯

^a^ Values of heartburn, reflux, and other symptoms from gastroesophageal reflux disease in dexlansoprazole group were lower compared to the control group. ^b^ Studies presented many methodological differences. ⨁⨁⨁◯—Moderate.

## Data Availability

Data are contained within the article.

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
