# Peer review of "The Effect of Dexlansoprazole on Gastroesophageal Reflux Disease: A Systematic Review and Meta-Analysis"

_ijms, 2024, doi:10.3390/ijms25021247_

Round 1

Reviewer 1 Report

Comments and Suggestions for Authors

The authors conducted an interesting systematic review and meta-analysis of dexlansoprazole's effect on gastroesophageal reflux disease with the evaluation of randomized clinical trail studies comparing dexlansoprazole to other proton pump inhibitors (PPIs) and placebo. The review content, tables, and figures were informative. The systematic review of dexlansoprazole compared to other PPIs in the literature will be of interest to the broad range of physicians treating patients suffering from the broad clinical spectrum of gastroesophageal reflux disease. I have some very minor comments to improve the manuscript.

1. The table columns need to be re-sized so that the individual words in the headings do not get split between multiple lines because some of the words on one line in the heading get one word divided between two lines, which can be distracting to readers.

2. Double check the spelling of words in the table because some of the words, even words in the column titles, occasionally have typographical errors where some of the words are misspelled.

Since this is a review paper, originality is really not a relevant component of the review of this paper. However, I do think that systematic reviews of the proton pump inhibitors on the market, like the current review, is relevant to the field and can be quite informative to clinicians who are trying to decide what PPI to use in their GERD patients.

This systematic review is a nice synthesis of recent clinical trials around dexlansoprazole and proton pump inhibitors. Although not absolutely necessary for this systematic review, discussion about the differences in the mechanisms of action of dexlansoprazole compared to some of the more common PPIs discussed could be informative.

Since this is a review paper, methodology is really not important. However, the methodology used by the authors is reasonable for the systematic review conducted.

The systematic review conclusion section is supported by the fairly balanced systemic review of the published clinical trials discussed in this review paper.

The listed references are appropriate.

The tables and figures are informative and support the discussion in the systematic review conducted. However, Figures 2 and 4 could go into Supplementary Materials since Figure 3 discusses 24-hour, heartburn-free days and Figure 4 discusses heartburn-free nights. Figures 3 and 4 do not both need to be in the main paper. Also, another table or figure summarizing the differences in the mechanisms of action of dexlansoprazole and the other common PPIs discussed could also be informative.

Overall, I think this systematic review is within the scope of the journal and can be of use to clinicians deciding on PPIs to use in their patients within the clinical spectrum of GERD.

Author Response

REVIEWER #1      

The authors conducted an interesting systematic review and meta-analysis of dexlansoprazole´s effect on gastroesophageal reflux disease with the evaluation of randomized clinical trail studies comparing dexlansoprazole to other proton pump inhibitors (PPIs) and placebo. The review content, tables, and figures were informative. The systematic review of dexlansoprazole compared to other PPIs in the literature will be of interest to the broad range of physicians treating patients suffering from the broad clinical spectrum of gastroesophageal reflux disease. I have some very minor comments to improve the manuscript.

  1. The table columns need to be re-sized so that the individual words in the headings do not get split between multiple lines because some of the words on one line in the heading get one word divided between two lines, which can be distracting to readers.

Author's response: The authors agree with the reviewer. Thus, the tables were configured as suggested.

  1. Double check the spelling of words in the table because some of the words, even words in

the column titles, occasionally have typographical errors where some of the words are

misspelled.

Author's response: The spelling of words in the table was checked and corrections were made.

  1. The tables and figures are informative and support the discussion in the systematic review conducted. However, Figures 2 and 4 could go into Supplementary Materials since Figure 3 discusses 24-hour, heartburn-free days and Figure 4 discusses heartburn-free nights. Figures 3 and 4 do not both need to be in the main paper. Also, another table or figure summarizing the differences in the mechanisms of action of dexlansoprazole and the other common PPIs discussed could also be informative.

Author's response: We understand the reviewer's comment, so Figure 2 was added to the supplementary material. Regarding Figure 4, it addresses the result of meta-analysis, as well as Figure 3. Thus, if the reviewer agrees, the authors consider it important to maintain quantitative results in the main manuscript. In addition, we appreciate the suggestion of the addition of more information regarding the mechanism of action of the pharmaceuticals, and we introduce a paragraph with this at the beginning of the Discussion (marked in blue).

Reviewer 2 Report

Comments and Suggestions for Authors

Dear Authors

This is an interesting work about the effects of dexlansoprazole in GERD.

I have some questions about this manuscript

What criteria were used to define GERD? Please clarify further in the text.

The articles are from very distant years and there have been changes in consensus.

Would this impact the analysis of the results?

24-hour Phmetry is the gold standard test for evaluating GERD. Why was it not considered in the selection of articles analyzed?

Table 2 must be placed horizontally

Comments on the Quality of English Language

The English is fine

Author Response

REVIEWER #2      

This is an interesting work about the effects of dexlansoprazole in GERD.

I have some questions about this manuscript

  1. What criteria were used to define GERD? Please clarify further in the text.

Authors’ response: The criteria to define GERD: beyond the clinical characteristics, with the presence of heartburn and acid regurgitation, GERD was proven by endoscopy for the assessment of eligibility.

  1. The articles are from very distant years and there have been changes in consensus. Would this impact the analysis of the results?

Author’s response:  We understand the reviewer's question. Based on the included articles, there was an 11-year amplitude from the first to the ultimate eligible study. Thus, the present systematic review aimed to assess whether there were differences in the clinical effect of the evaluated drug. Although other PPIs have been described as effective, there is still no consensus on which therapy is most effective. Although some studies already have more than 10 years of publication, and despite the limitations already expressed, we reinforced that the RCT evaluations have been performed as congruent as possible with the latest consensus reported.

  1. 24-hour Phmetry is the gold standard test for evaluating GERD. Why was it not considered in the selection of articles analyzed?

Author’s response:  The authors understand the reviewer's comment, really the pHmetry is the most accurate assessment. However, it is not the initial exam for GERD. Evaluation by endoscopy, besides being effective, is primarily advocated to discard the possibility of malignancy and disease by H Pylori. In addition, pHmetry is an exam with less availability in the clinical routine, which possibly explains its smallest presence in clinical trials. We agree with the relevant questioning of the reviewer, given that we highlight in the discussion section the limitations of the studies included concerning the topic addressed by the reviewer.

  1. Table 2 must be placed horizontally

Author's response: As proposed by the reviewer the table was adjusted horizontally.